# Unsupervised learning of multimodal image registration using domain adaptation with projected Earth Mover's discrepancies

**Mattias P. Heinrich**                                      HEINRICH@IMI.UNI-LUEBECK.DE

**Lasse Hansen**                                          HANSEN@IMI.UNI-LUEBECK.DE

*Institute of Medical Informatics, Universität zu Lübeck, Lübeck, Germany*

## Abstract

Multimodal image registration is a very challenging problem for deep learning approaches. Most current work focuses on either supervised learning that requires labelled training scans and may yield models that bias towards annotated structures or unsupervised approaches that are based on hand-crafted similarity metrics and may therefore not outperform their classical non-trained counterparts. We believe that unsupervised domain adaptation can be beneficial in overcoming the current limitations for multimodal registration, where good metrics are hard to define. Domain adaptation has so far been mainly limited to classification problems. We propose the first use of unsupervised domain adaptation for discrete multimodal registration. Based on a source domain for which quantised displacement labels are available as supervision, we transfer the output distribution of the network to better resemble the target domain (other modality) using classifier discrepancies. To improve upon the sliced Wasserstein metric for 2D histograms, we present a novel approximation that projects predictions into 1D and computes the L1 distance of their cumulative sums. Our proof-of-concept demonstrates the applicability of domain transfer from mono- to multimodal (multi-contrast) 2D registration of canine MRI scans and improves the registration accuracy from 33% (using sliced Wasserstein) to 44%.

**Keywords:** Multi-modal registration, domain adaptation, discrete displacements

## 1. Introduction

Gathering labelled training data for learning-based multimodal registration is very time-consuming and expensive. To train supervise methods either a large number of corresponding landmarks (cf. (Xiao et al., 2019)) or detailed anatomical multi-label segmentations are required (cf. (Hu et al., 2018)), which often cause bias or under-coverage. To circumvent the need for corresponding labels in multimodal / multi-domain images, unsupervised domain adaptation based on classifier discrepancies has been popularised in computer vision for classification and segmentation tasks e.g. in (Lee et al., 2019). Variants of discrepancy measures include e.g. the Earth Mover's distance (EMD) (Werman et al., 1985) for 1D cases and specialised solutions for 2D histograms in (Ling and Okada, 2006), but they are in general computationally expensive, approximative or based on sensitive hyperparameters.

**Contribution:** We are the first to propose domain adaptation for medical registration and adapt the task to a discrete displacement labelling. Using the maximum classifier

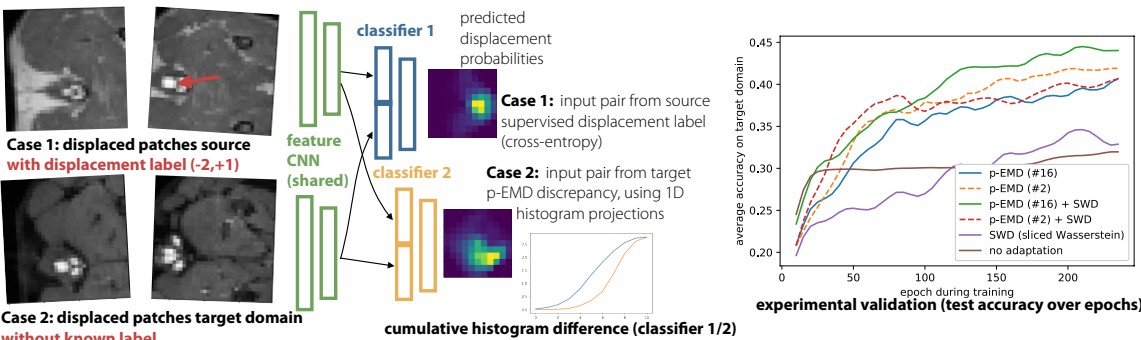

Figure 1: Our method comprises a shared feature network and two classifiers for maximum discrepancy domain adaption. The results demonstrate the superiority of our new p-EMD metric (44% vs 33% accuracy) compared to sliced Wasserstein (SWD).

discrepancy approach (Saito et al., 2018) together with a novel 2D histogram Earth Movers distance, we substantially improve over the sliced Wasserstein metric (Lee et al., 2019).

**Related Work:** Recent methods for supervised learning of multimodal registration include (Simonovsky et al., 2016), who use a twin CNN architecture to learn the similarity of patches using aligned multi-modal training data. (Hu et al., 2018) and (Hering et al., 2019), both use anatomical segmentations to train a U-net like registration network, while the latter add a normalised gradient metric. The use of discrete displacements in deep learning based registration was proposed in (Heinrich, 2019) to capture large deformations. Unpaired unsupervised learning for multi-modal medical images has so far been restricted to modality synthesis using e.g. Cycle-GANs in (Wolterink et al., 2017). Very recent methods have shown promise for unsupervised domain adaptation and knowledge distillation for medical image classification and multimodal segmentation (Dou et al., 2020).

## 2. Methods and Material:

Unsupervised domain adaption has so far mainly shown success for classification tasks. We hence adapt the task of multimodal image registration to a discrete labelling problem, similar as done in (Heinrich, 2019). Here, we restrict ourselves to 2D patch based registration to demonstrate a proof-of-concept. For training, we extract large patches with a random offset within a grid of 5x5 discrete displacements to pose registration as a 25-class classification problem. We add 3D affine augmentations to avoid trivial overlap within two patches.

During training we have access to a labelled source domain dataset (in this case MRI T1 patches) with known displacements and an unlabelled target domain dataset (MRI T2 patches). For feature extraction, we use a feed-forward net comprising four blocks of Conv2d, InstanceNorm and PReLU (13k weights) within a twin architecture that shares weights across both patches. This feature network produces a 18x18 map with 16 channels. Subsequently, we concatenate both patches and feed them into a three block classification network (70k weights) that predicts a 25D classification vector (encoding the displacements).

We employ the maximum discrepancy of classifiers approach of (Saito et al., 2018), which uses two similar classifiers (with different random initialisation). The training alternates between the following three steps (see also Figure 1): 1) optimise features and classifier on labelled source data, 2) maximise discrepancy measure of both classifiers on target domain while freezing the feature weights and minimising the classification loss on the labelled source data, 3) minimise the discrepancy measure of classifiers while updating only the feature weights. This process helps to identify target samples outside the classifier's support region in step 2 and brings the target feature distributions closer to the source ones during step 3. Step 3 is repeated twice as proposed in (Saito et al., 2018). We make two modifications that greatly improve stability: 1) we only update the first classifier in step 1 to avoid overfitting (too similar decision boundaries) on the source domain before the domain adaptation begins to improve, 2) we use the cross entropy loss for classifiers, but scale the predictions by 0.1 before computing the softmax output for the discrepancy measures to reduce overly confident predictions as motivated by (Kuleshov et al., 2018).

**Fast projected Earth Mover's distance (p-EMD) for multidimensional histograms:** A disadvantage of the sliced Wasserstein distance (Lee et al., 2019) for our application is its invariance to permutations of histogram bins / classes. This may be beneficial when no natural measure of class proximity exists. Yet, in our case the prediction can be converted into a 2D spatial probability map for x- and y-displacements. We therefore propose a new approximate metric for higher-order histograms (p-EMD) that takes these specificities into account. It is faster and easier to differentiate than conventional algorithms. Given that the distributions are close to monomodal Gaussians and based on the fact that exact algorithms for computing EMD for normalised 1D histograms in linear complexity exist (Werman et al., 1985), we approximate the optimal transport cost by projecting the (softmax) normalised 2D histograms onto a number of rotated lines (we use either 2 or 16 projections with angles between 0 and 90 degrees and use bilinear interpolation). We then employ the L1 distance of their cumulative sums to compute the p-EMD and average the values across projections (see Figure 1), this correlates nearly perfectly with exhaustive EMD computations and is much more stable in our experiments than the 2D diffusion distance of (Ling and Okada, 2006).

## 3. Results and Discussion

We created a multimodal dataset for patch registration based on 9 3D T1 and T2 MRI scans of canine legs as provided by the 2013 MICCAI SATA challenge (Asman et al., 2013) with 5120 patch pairs in each modality. T1to T2 MRI is a simpler domain adaptation task, we thus increase the complexity by applying slightly different normalisations to the patches (global mean and variance for T1, and patch-wise for T2). The range of displacements was $\{-38, -19, 0, +19, +38\}^2$ pixels (posing a very challenging large motion problem) and each patch comprises a region of 77x77 voxels downsampled to half resolution. The supervised training was restricted to monomodal data (T1→T1), while the multi-modal tests were performed on T2→T1, T1→T2 and T2→T2. The average accuracy (prediction of 1 of the 25 classes) across 5 runs is shown in Fig. 1 (right), yielding only a modest improvement from 31.9% (no adaptation) for sliced Wasserstein (SWD) to 33.2%. Both of our variants p-EMD (2 or 16 projections) reach accuracies over 40%, adding the losses of p-EMD (#16)

and SWD is best with 44.1% (see also Table 1). Future work will focus on more elaborate experiments and evaluation, e.g. integrating the patch-wise displacement estimation into global transformation models (e.g. using the instance optimisation proposed in (Heinrich, 2019)), extending it to 3D and comparison to classical multimodal metrics.

## Acknowledgments

This work was in part supported by the German ministry of Education and Research (BMBF) within the project Multi-Task Deep Learning for Large-Scale Multimodal Biomedical Image Analysis (MDLMA) FKZ 031L0202B.

Table 1: Overview of label accuracy results for multi-modal MRI registrations

| method | registration label accuracy |
|---|---|
| no registration (guessing) | 4.0% |
| no adaptation (training on T1) | 31.9% |
| sliced Wasserstein (SWD) | 33.2% |
| p-EMD (#16) and SWD (ours) | 44.1% |

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
