# OpenReview forum: "Unsupervised learning of multimodal image registration using domain adaptation with projected Earth Mover’s discrepancies"
_MIDL.io/2020/Conference — MIDL 2020_

### Official Review · AnonReviewer3 · 2020-03-14
**Unsupervised domain adaptation for registration using a 2D histogram distance (Earth Mover)**

**Rating:** 3
**Confidence:** 4

**Review:**

This paper proposes an interesting and still new application of domain adaptation, image registration. The purpose is to adapt a network trained to estimate the displacement on MRI T1 patches to T2 patches. The proposed network has access to pairs of patches in the source domain, with their displacement label. In the target domain, it only has access to displaced patches. The idea to make displacement "similar" in the source and target domain is to match the histograms of displacements. For this end, a projected Earth Mover’s distance metric is proposed and compared to Wasserstein distance.

The ideas our straightforward. Nonetheless, a more friendly introduction to histogram distances could have been proposed. For example, it could have been beneficial to show histograms, and the resulting Wassertein distance versus the proposed one.
A results table comparing baseline without adaptation, Wassertein, and proposed method could have been produced for clarity. The results/ discussion section is limited, so the paper could be better organized.

Overall the idea is nice and the application still new.

---

### Official Review · AnonReviewer2 · 2020-03-19
**Unsupervised multimodal discrete registration using domain adaptation**

**Rating:** 2
**Confidence:** 3

**Review:**

The approach views registration as a discretized multilabel classification task, and exploits the maximum classifier discrepancy (MCD) idea from the field of domain adaptation. This allows to exploit annotations on a given modality to train a registration network on other target domains.
The main contribution is in the way the discrepancy measure is set, using 1d projections of the 2d histogram of displacement label probabilities to both preserve spatial information and retain ease of computation. The approach is demonstrated on 2D patches.

- What is the motivation for domain adaptation? "Gathering labelled training data for learning-based multimodal registration is very time-consuming and expensive" True, but the application chosen here doesn't really relate to this point. Also, T1/T2 to illustrate domain adaptation is a bit stretched as this can be performed with standard metrics (NMI, etc.)

- Along the same line of thought, the approach restricts to the use of a shared feature CNN between source and target domains, which may limit how different the domain appearances can be; empirical validation in a truly multimodal setting would be useful here (alternatively showing improvements over standard un/supervised registration with multimodal image similarity metrics).

- If only one result is reported regarding accuracy, it should be directly in terms of displacement error rather than label accuracy. The latter only relates to the specific choice of formulation and is tough to interpret.

---

### Official Review · AnonReviewer5 · 2020-03-19
**Well grounded exploratory work**

**Rating:** 3
**Confidence:** 3

**Review:**

This paper describes a domain adaptation-based unsupervised learning of medical image registration. Preliminary results show that coarse patch-based displacement classification can be performed well using domain adaptation-based unsupervised learning, and show improvement over traditional methods. The paper is well written and concepts are explained as well as they can be in the limited 3-page space.

Pros: Use of deep learning concepts are well justified in the paper. Every decision comes with an explanation, whether it the type of loss function used, they way weights are updated to prevent over-fitting, or the way predictions are scaled (explained with citation).  This is a welcome change from reading several deep learning papers that simply use some network architecture, loss functions, parameters, etc. without explanation.

Cons: The accuracy numbers are low for all the reported methods. It would be nice to intuitively understand how the accuracy numbers translate to actual registration errors. This may be hard to compute but authors could use something like SSD, for instance, to understand what kind of registration errors are produced when the accuracy of the network is ~40%.

---

### Official Review · AnonReviewer4 · 2020-03-19
**A mono- and multi- modal approach for image registration**

**Rating:** 2
**Confidence:** 3

**Review:**

The paper is particularly hard to read and with a poor English. Authors must correct the grammar and spelling mistakes if the paper is accepted.

Pros:
- challenging and interesting problem: multi-modal registration with deep learning
- combining domain adaptation and registration is a novel idea
- explores some interesting concepts such as approximating the Earth Mover's distance via a 1D projection

Cons:
- 2D approach. Seems hard to extend to large 3D volumes
- poor English
- experiments are unclear. The registration is not directly evaluated. In particular, are the registration outputs well regularised?

---

### Meta-Review · Area_Chair1 · 2020-04-06
**MetaReview of Paper131 by AreaChair1**

**Rating:** 3

**Metareview:**

Two reviewers recommend weak acceptance while the other two reviewers recommend weak rejection. However, most of them seem to acknowledge the novelty of the paper. Since this is a short paper submission which seems to introduce a novel idea, I'm therefore inclined for accepting this publication.

Importantly, for the camera ready version, please take into account the comments about the poor english quality raised by R4 and those related to how the organization of the paper could be improved made by R3.  I suggest the authors to ask a native english speaker or an expert to proofread their manuscript.

**Paper Type:**

methodological development

---

### Decision · Program_Chairs · 2020-04-11

Accept